# ROYAL SOCIETY
# OPEN SCIENCE

analysis/mathematical physics/materials science

quasi-Herglotz functions, non-passive systems, approximation, convex optimization, sum rules

**Author for correspondence:**
Y. Ivanenko
e-mail: yevhen.ivanenko@lnu.se

# Quasi-Herglotz functions and convex optimization

Y. Ivanenko[1], M. Nedic[2], M. Gustafsson[3], B. L. G. Jonsson[4], A. Luger[2] and S. Nordebo[1]

[1]Department of Physics and Electrical Engineering, Linnæus University, 351 95 Växjö, Sweden
[2]Department of Mathematics, Stockholm University, 106 91 Stockholm, Sweden
[3]Department of Electrical and Information Technology, Lund University, Box 118, 221 00 Lund, Sweden
[4]School of Electrical Engineering and Computer Science, KTH Royal Institute of Technology, 100 44 Stockholm, Sweden

YI, 0000-0002-3928-6064; MN, 0000-0001-7867-5874;
MG, 0000-0003-4362-5716; BLGJ, 0000-0001-7269-5241;
SN, 0000-0002-7018-6248

We introduce the set of quasi-Herglotz functions and demonstrate that it has properties useful in the modelling of non-passive systems. The linear space of quasi-Herglotz functions constitutes a natural extension of the convex cone of Herglotz functions. It consists of differences of Herglotz functions and we show that several of the important properties and modelling perspectives are inherited by the new set of quasi-Herglotz functions. In particular, this applies to their integral representations, the associated integral identities or sum rules (with adequate additional assumptions), their boundary values on the real axis and the associated approximation theory. Numerical examples are included to demonstrate the modelling of a non-passive gain medium formulated as a convex optimization problem, where the generating measure is modelled by using a finite expansion of B-splines and point masses.

## 1. Introduction

It is well known that an *admittance passive system* (admittance, impedance, electromagnetic constitutive relations, etc.), i.e. a system that absorbs more energy than it emits [1], can be represented mathematically by a symmetric Herglotz function (or positive real function) (e.g. [2–7]). The condition of passivity implies, among other things, that the system also has to be causal [8]. Furthermore, the integral representation formula for symmetric Herglotz functions leads to integral identities or sum rules [4,6] that are useful to derive physical bounds in a variety of technical applications such as radar absorbers [9], passive metamaterials [10], high-impedance surfaces [11], antennas [12,13], reflection coefficients [14], waveguides [15] and periodic structures [16], only to mention a few. The integral

representation formula can also be used in a convex optimization setting to construct an optimal approximating passive realization of a desired target response [17,18], which is typically given on a finite closed interval of the real (frequency) axis. Optimal realizations of passive metamaterials are typical examples, where it is, for example, desired to synthesize low-loss materials with negative refractive index over a frequency interval [10,17,18]. However, there exist many practically important systems that are causal, but not passive, and thus we introduce a new class of functions to model them.

As a motivation for the study of the new class of functions, we refer to the use of gain media which has been proposed to improve the light localization effects in plasmonics, with applications such as plasmon waveguides, extraordinary transmission, perfect lenses, artificial magnetism, negative refractive index, cloaking, tunnelling, high directivity radiators, optical nanocircuits, nanowires etc. (e.g. [19–25] and references therein). Here, the use of gain media refers to the use of fluorescent dyes through optical pumping for which there exist explicit Lorentz type of resonance models in the standard laser literature (e.g. [21,25–28] and references therein). Hence, laser pumping is a physical mechanism that allows for a linearized description of the medium in terms of a dielectric permittivity that can have a negative imaginary part over some frequency intervals. Naturally, it has been recognized that such models must satisfy causality and the associated Kramers–Kronig relations [29]. However, our purpose of employing the new class of functions in this context is to add the restrictions imposed by passivity outside the (non-passive) emitting frequency range for determination of an optimal realization of a non-passive medium characterized by a permittivity function. It is emphasized that the term *realizability* is employed here in the sense of realizability theory as in [30]. This means that a given system response is realizable, not as a physical system, but rather as a function possessing mathematically well defined properties of physical significance, such as causality and passivity [30], and also as in our case, having some *a priori* assumed regularity properties regarding its boundary values on the real (frequency) axis.

The boundary values of analytic functions representing causal systems are treated classically in $L^2$ spaces as in Titchmarsh's theorem [5,29] or in the sense of tempered distributions as in [2,31]. There are a few results concerned with approximation theory or interpolation problems associated with partial information on the real axis (or on the unit circle). For example, bounds on the dispersion for finite-frequency-range Kramers–Kronig relations based on Stieltjes functions are presented in [32], and in [33] an approximation theory is given with density results for Hardy space approximants targeted for $L^p$ functions defined on subsets of the circle. Furthermore, a related bounded extremal problem is examined in [34] with point-wise constraints on the complementary part of the circle.

In this paper, we are interested in extending the class of admittance passive systems to include certain causal, non-passive systems. This extension is aimed to preserve the integral representation formula for the system, as well as, in certain cases, a sum rule. As a characterization of the full class of functions that satisfy the sum-rule identities seems out of reach, we use a class of functions that includes all Herglotz functions, and for which the sum-rule identities still hold under some appropriate additional assumptions regarding their asymptotic expansions. Moreover, it is also desirable that the new class of functions can be incorporated in an approximation theory, similar to the one for Herglotz functions [18]. It turns out that differences of Herglotz functions are suitable in this sense and we define the (real) vector space generated by Herglotz functions as the space of *quasi-Herglotz functions*. As for the approximation theory, we follow a slightly different route from [33,34] and consider, as approximants, certain subspaces of quasi-Herglotz functions which are Hölder continuously extendable to a neighbourhood of a given approximation interval on the real line, equipped with the topology from a larger $L^p$ space. This is a formulation that will imply that even smaller subspaces generated by finite B-spline expansions (useful in convex optimization) will be dense in the larger set of approximants. Numerical examples are included to demonstrate the approximation approach by modelling of a given non-passive system by solving a convex optimization problem. Here, the generating measure is modelled by using a finite expansion of B-splines and point masses.

The rest of the paper is organized as follows. In §2, we introduce the set of quasi-Herglotz functions and discuss their basic properties, integral representations and boundary values. In §3, the sum rules are formulated and proved. In §4, the mathematical approximation theory and related convex optimization are formulated. It is based on certain assumptions regarding the Hölder continuity of the approximating quasi-Herglotz functions extended to the real line. In §5, the theory is illustrated by numerical examples, and the paper ends with conclusions in §6.

## 2. Quasi-Herglotz functions

### 2.1. Background

An important function class in applied mathematics is the class of so-called *Herglotz functions*. Known also under a variety of different names, such as Herglotz–Nevanlinna functions, Pick functions and R-functions, these are analytic functions on the upper half-plane

$$\mathbb{C}^+ := \{z := x + iy \in \mathbb{C} \,|\, y > 0\} \tag{2.1}$$

having non-negative imaginary part [3,4]. A major significance of this class of functions lies in the fact that the subclass of all symmetric Herglotz functions, i.e. Herglotz functions with the property

$$h(z) = -h(-z^*)^*, \tag{2.2}$$

is closely connected with passive systems [2]. Above, the superscript $(\cdot)^*$ denotes complex conjugation.

One of the most powerful tools in the theory of Herglotz functions is the existence of an integral representation formula [3,4]. This well-known formula states that a function $h\colon \mathbb{C}^+ \to \mathbb{C}$ is a Herglotz function if and only if it can be written, for any $z \in \mathbb{C}^+$, as

$$h(z) = a_+ + b_+ z + \int_{\mathbb{R}} \frac{1 + \xi z}{\xi - z} \, d\sigma_+(\xi), \tag{2.3}$$

where $a_+ \in \mathbb{R}$, $b_+ \geq 0$ and $\sigma_+$ is a finite positive Borel measure on $\mathbb{R}$, and the subscrpit $(\cdot)_+$ is used to highlight the fact that these parameters represent a function with *non-negative* imaginary part. Furthermore, the correspondence between the function $h$ and the triple of its representing parameters $(a_+, b_+, \sigma_+)$ is unique.

If we are, instead, considering a symmetric Herglotz function, condition (2.2) implies first that the function must take purely imaginary values along the imaginary axis, yielding that the coefficient $a_+$ from representation (2.3) must be zero. Furthermore, the Stieltjes inversion formula [3] implies that the measure $\sigma_+$ from representation (2.3) must be even, i.e. $\sigma_+(U) = \sigma_+(-U)$ for any Borel measurable set $U \subseteq \mathbb{R}$, where $-U := \{x \in \mathbb{R} \,|\, -x \in U\}$.

As such, all symmetric Herglotz functions $h$ admit, for $z \in \mathbb{C}^+$, an integral representation of the form

$$h(z) = b_+ z + \text{p.v.} \int_{\mathbb{R}} \frac{1 + \xi^2}{\xi - z} \, d\sigma_+(\xi), \tag{2.4}$$

where $b_+$ and $\sigma_+$ are as in representation (2.3), with the additional constraint that the measure $\sigma_+$ is symmetric (cf. [2–5,7]) and p.v. denotes that the integral in representation (2.4) is taken as the Cauchy principal value at infinity. Observe that it is necessary to view the above integral in the principal value sense to ensure convergence. Indeed, for any fixed $z \in \mathbb{C}^+$, the integrand grows linearly at $\pm\infty$ and is, hence, not necessarily integrable with respect to the measure $\sigma_+$. Note, furthermore, that this is not the case in representation (2.3), where the integrand is a bounded function on $\mathbb{R}$ for any fixed $z \in \mathbb{C}^+$.

### 2.2. Basic properties

We now introduce the following class of analytic functions on the upper half-plane.

**Definition 2.1** An analytic function $q\colon \mathbb{C}^+ \to \mathbb{C}$ is called a *quasi-Herglotz function* if there exist two Herglotz functions $h_1$ and $h_2$, such that

$$q(z) = h_1(z) - h_2(z) \tag{2.5}$$

for any $z \in \mathbb{C}^+$. Analogously, an analytic function $q\colon \mathbb{C}^+ \to \mathbb{C}$ is called a *symmetric quasi-Herglotz function* if there exist two symmetric Herglotz functions $h_1$ and $h_2$, such that equality (2.5) holds for all $z \in \mathbb{C}^+$. The set of all quasi-Herglotz functions is denoted by $\mathcal{Q}$, while the set of all symmetric quasi-Herglotz functions is denoted by $\mathcal{Q}_{\text{sym}}$.

We mention two trivial observations. First, any Herglotz (resp. symmetric Herglotz) function is also a quasi-Herglotz (resp. symmetric quasi-Herglotz) function, as we only need to take the function $h_2$ in definition 2.1 to be identically equal to zero. Second, there is an element of non-uniqueness in definition 2.1. If an analytic function $q$ can be written as in formula (2.5) for some Herglotz (resp.

symmetric Herglotz) functions $h_1$ and $h_2$, then it can also be written as

$$q(z) = (h_1 + h_3)(z) - (h_2 + h_3)(z), \tag{2.6}$$

where $z \in \mathbb{C}^+$, for any other Herglotz (resp. symmetric Herglotz) function $h_3$.

## 2.3. Integral representations

It is an immediate consequence of the integral representation formulae (2.3) and (2.4) that quasi-Herglotz functions in the sets $\mathcal{Q}$ and $\mathcal{Q}_{\mathrm{sym}}$ admit similar integral representations. Any function $q \in \mathcal{Q}$ can be written, for $z \in \mathbb{C}^+$, as

$$q(z) = a + bz + \int_{\mathbb{R}} \frac{1 + \xi z}{\xi - z} \, \mathrm{d}\sigma(\xi), \tag{2.7}$$

where $a$ and $b$ are real numbers and $\sigma$ is a signed Borel measure. In particular, if $q$ is given as $q = h_1 - h_2$, then $a = a_{+,1} - a_{+,2}$, $b = b_{+,1} - b_{+,2}$ and $\sigma = \sigma_{+,1} - \sigma_{+,2}$ where, for $j = 1, 2$, the parameters $a_{+,j}$, $b_{+,j}$ and $\sigma_{+,j}$ are the representing parameters for the Herglotz function $h_j$ in the sense of representation (2.3).

Similarly, any function $q = h_1 - h_2$ in the class $\mathcal{Q}_{\mathrm{sym}}$ can be written, for $z \in \mathbb{C}^+$, as

$$q(z) = bz + \mathrm{p.v.} \int_{\mathbb{R}} \frac{1 + \xi^2}{\xi - z} \, \mathrm{d}\sigma(\xi), \tag{2.8}$$

where $b$ and $\sigma$ are as in the previous case.

Note that, despite the element of non-uniqueness in definition 2.1 discussed in §2.2, the triple of representing parameters $(a, b, \sigma)$ corresponding to a quasi-Herglotz function $q$ in the sense of representation (2.7) is determined uniquely by the function $q$.

The integral representation formula (2.3) for ordinary Herglotz functions may also be written in terms of a not necessarily finite measure $\beta_+$. Indeed, one can show that the right-hand side of representation (2.3) may equivalently be written as

$$a_+ + b_+ z + \int_{\mathbb{R}} \left( \frac{1}{\xi - z} - \frac{\xi}{1 + \xi^2} \right) \mathrm{d}\beta_+(\xi), \tag{2.9}$$

where $a_+$ and $b_+$ are as before and $\beta_+$ is a positive Borel measure on $\mathbb{R}$ satisfying the growth condition

$$\int_{\mathbb{R}} \frac{1}{1 + \xi^2} \, \mathrm{d}\beta_+(\xi) < \infty. \tag{2.10}$$

However, an integral representation of this form cannot yield an integral representation for all quasi-Herglotz functions, as the difference of two measures satisfying the growth condition (2.10) is not necessarily well-defined. Nevertheless, some quasi-Herglotz functions $q$ do admit an integral representation of the form

$$q(z) = a + bz + \int_{\mathbb{R}} \left( \frac{1}{\xi - z} - \frac{\xi}{1 + \xi^2} \right) \mathrm{d}\beta(\xi), \tag{2.11}$$

and one case where this happens, which will appear later in §§4 and 5, is when the measure $\sigma$ from representation (2.7) has compact support. Then, the measure $\beta$ in representation (2.11) may be defined via $\mathrm{d}\beta(\xi) = (1 + \xi^2) \, \mathrm{d}\sigma(\xi)$.

## 2.4. Boundary values

In general, Herglotz functions, as well as quasi-Herglotz functions, and in particular their imaginary parts, have boundary values (on the real line) only in the distributional sense (e.g. [2,6,18,31]). In what follows, however, we will be interested in complex-valued functions on some interval $\Omega \subset \mathbb{R}$ which appear as continuous extensions of suitable quasi-Herglotz functions.

First, we want to mention certain inclusions of function spaces, which will be very useful in §4. As usual, we let $C(\Omega)$ denote the Banach space consisting of all complex-valued continuous functions defined on some compact interval $\Omega \subset \mathbb{R}$ equipped with the standard max-norm $\| \cdot \|_{\infty}$. The Hölder space with exponent $0 < \alpha < 1$ is denoted $C^{0,\alpha}(\Omega)$ and the corresponding norm is denoted $\| \cdot \|_{\alpha}$ (cf. [35, pp. 94–104]). Further, let $\mathrm{L}^p(w, \Omega)$ denote the Banach space with norm $\|f\|_{\mathrm{L}^p(w,\Omega)} = ( \int_{\Omega} w(x) |f(x)|^p \, \mathrm{d}x )^{1/p}$, where $1 \leq p < \infty$ and $w > 0$ denotes a positive continuous weight function on $\Omega$ (cf. [36]). The Banach

space $L^\infty(w, \Omega)$ is similarly equipped with the norm $\|f\|_{L^\infty(w,\Omega)}$ defined by taking the essential supremum [36] of the function $w|f|$. Then, the spaces defined above satisfy the following inclusions:

$$C^{0,\alpha}(\Omega) \subset C(\Omega) \subset L^p(w, \Omega), \tag{2.12}$$

where $0 < \alpha < 1$ and $1 \le p \le \infty$.

Second, recall that the property that assures the existence of boundary values of quasi-Herglotz functions (i.e. of both the real and imaginary parts) is Hölder continuity of the density of the measure. More precisely, the following theorem holds (see [18, Thm. 2.2] for the argument).

**Theorem 2.2** *Let $q$ be a quasi-Herglotz function with representing parameters $(a, b, \sigma)$ and let $\Omega \subset \mathbb{R}$ be a compact interval. Then the function $q$ can be Hölder continuously (with Hölder exponent $\alpha$) extended to $\Omega \cup \mathbb{C}^+$ if and only if the measure $\sigma$ is absolutely continuous on the closure of some open neighbourhood $\mathcal{O}$ of $\Omega$ and the corresponding restriction $\sigma|_{\overline{\mathcal{O}}}$ has a Hölder continuous density $\sigma'_\Omega$ (with Hölder exponent $\alpha$), i.e. it belongs to the space $C^{0,\alpha}(\overline{\mathcal{O}})$. In this case, for every $x \in \Omega$, this extension is given by*

$$q(x) = a + bx + \text{p.v.} \int_{\mathbb{R}} \frac{1 + \xi x}{\xi - x} \, d\sigma(\xi) + i\pi(1 + x^2)\sigma'_\Omega(x), \tag{2.13}$$

*where the integral is taken as a Cauchy principal value both at infinity and at the singularity $x \in \mathbb{R}$.*

# 3. Sum rules

One of the most important properties of Herglotz functions are the, so-called, *sum-rule identities* [6, Thm. 4.1] and [4]. These identities relate weighted integrals of the imaginary part of a Herglotz function, via the moments of its representing measure, to the coefficients of the asymptotic expansion of the function at the points zero and infinity.

The asymptotic expansions we are interested in are always taken with respect to non-tangential limits in a Stoltz domain. A Stoltz domain with parameter $\theta \in (0, \pi/2]$ is the angular domain

$$\{z \in \mathbb{C}^+ \mid \theta < \text{Arg}(z) < \pi - \theta\}. \tag{3.1}$$

As such, the limit $z \hat{\to} 0$ (resp. $z \hat{\to} \infty$) denotes that the limit $|z| \to 0$ (resp. $|z| \to \infty$) is taken in any Stoltz domain as above.

Consider now the following definitions.

**Definition 3.1** Let $q$ be a quasi-Herglotz function. We say that $q$ admits, at $z = 0$, an *asymptotic expansion of order $M \ge -1$* if there exist real numbers $a_{-1}, a_0, a_1, \ldots, a_M$ such that $q$ can be written as

$$q(z) = \frac{a_{-1}}{z} + a_0 + a_1 z + \ldots + a_M z^M + o(z^M) \quad \text{as } z \hat{\to} 0. \tag{3.2}$$

**Definition 3.2** Let $q$ be a quasi-Herglotz function. We say that $q$ admits, at $z = \infty$, an *asymptotic expansion of order $K \ge -1$* if there exist real numbers $b_1, b_0, b_{-1}, \ldots, b_{-K}$ such that $q$ can be written as

$$q(z) = b_1 z + b_0 + \frac{b_{-1}}{z} + \ldots + \frac{b_{-K}}{z^K} + o\left(\frac{1}{z^K}\right) \quad \text{as } z \hat{\to} \infty. \tag{3.3}$$

At $z = 0$, an expansion of order $M = -1$ always exists for any quasi-Herglotz function $q$, as it always exists for any two Herglotz functions $h_1$ and $h_2$, cf. [3,6], yielding that

$$\lim_{z \hat{\to} 0} z q(z) = -\sigma(\{0\}), \tag{3.4}$$

where the signed measure $\sigma$ is as in representation (2.7). Similarly, at $z = \infty$, an expansion of order $K = -1$ always exists for any quasi-Herglotz function $q$, as it always exists for any two Herglotz functions $h_1$ and $h_2$, cf. [3,6], yielding that

$$\lim_{z \hat{\to} \infty} \frac{q(z)}{z} = b, \tag{3.5}$$

where the number $b$ is as in representation (2.7). Furthermore, the number $b$ equals the number $b_1$ appearing in definition 3.2.

We may now derive the following sum-rule theorem.

**Theorem 3.3** *The following two statements hold.*

(i) *Let $q = h_1 - h_2$ be a quasi-Herglotz function, such that at least one of the Herglotz functions $h_1$ and $h_2$ admits, at $z = 0$, an asymptotic expansion (3.2) of some order $M \geq -1$. Then, for some integer $N_0 \geq 1$ with $2N_0 - 1 \leq M$, the limit*

$$\lim_{\varepsilon \to 0^+} \lim_{y \to 0^+} \int_{\varepsilon < |x| < (1/\varepsilon)} x^{-2N_0} \operatorname{Im}\{q(x + iy)\} \, dx \tag{3.6}$$

*exists as a finite number if and only if the function $q$ admits, at $z = 0$, an asymptotic expansion (3.2) of order $2N_0 - 1$.*

(ii) *Let $q = h_1 - h_2$ be a quasi-Herglotz function, such that at least one of the Herglotz functions $h_1$ and $h_2$ admits, at $z = \infty$, an asymptotic expansion (3.3) of some order $K \geq -1$. Then, for some integer $N_\infty \geq 0$ with $2N_\infty + 1 \leq K$, the limit*

$$\lim_{\varepsilon \to 0^+} \lim_{y \to 0^+} \int_{\varepsilon < |x| < (1/\varepsilon)} x^{2N_\infty} \operatorname{Im}\{q(x + iy)\} \, dx \tag{3.7}$$

*exists as a finite number if and only if the function $q$ admits, at $z = \infty$, an asymptotic expansion (3.3) of order $2N_\infty + 1$.*

*Furthermore, the identities*

$$\lim_{\varepsilon \to 0^+} \lim_{y \to 0^+} \frac{1}{\pi} \int_{\varepsilon < |x| < (1/\varepsilon)} x^k \operatorname{Im}\{q(x + iy)\} \, dx = \begin{cases} a_{-k-1}, & -2N_0 \leq k \leq -3, \\ a_{-k-1} - b_{-k-1}, & -2 \leq k \leq 0, \\ -b_{-k-1}, & 1 \leq k \leq 2N_\infty \end{cases} \tag{3.8}$$

*are valid*

— *for $k = -2N_0, -2N_0 + 1, \ldots, -2$ if there exists an integer $N_0$ satisfying statement (i),*
— *for $k = 0, 1, \ldots, 2N_\infty$ if there exists an integer $N_\infty$ satisfying statement (ii),*
— *for $k = -1$ if there exist integers $N_0$ and $N_\infty$ satisfying statements (i) and (ii), respectively.*

*In formula (3.8), the numbers $a_{-1}, a_0, a_1, \ldots, a_{2N_0-1}$ are as in definition 3.1 and the numbers $b_{-1}, b_{-2}, \ldots, b_{-(2N_\infty+1)}$ are as in definition 3.2.*

*Proof.* In the case of statement (i), we may, without loss of generality, assume that, if we write $q = h_1 - h_2$, it is the function $h_2$ that admits, at $z = 0$, an asymptotic expansion (3.2) of some order $M \geq -1$.

Then, it follows from e.g. [6, Thm. 4.1], that the limit (3.6) for the function $h_2$ exists and, moreover, that the sum rules identities (3.8) hold for the function $h_2$ for all $k$ between $-M - 1$ and $-2$. Thus, the existence of the limit (3.6) for the function $q = h_1 - h_2$ is equivalent to the existence of the limit

$$\lim_{\varepsilon \to 0^+} \lim_{y \to 0^+} \int_{\varepsilon < |x| < (1/\varepsilon)} x^{-2N_0} \operatorname{Im}\{h_1(x + iy)\} \, dx, \tag{3.9}$$

and the existence of an asymptotic expansion of the function $q$ of the form (3.2) is equivalent to the existence of an analogous expansion of the function $h_1$. Statement (i) is then established by applying the sum rule for the function $h_1$, namely [6, Thm. 4.1].

The proof of statement (ii) follows an analogous reasoning. ∎

**Remark 3.4** For $q = h_1 - h_2$, the requirement of statement (i) in theorem 3.3 will certainly be satisfied if the representing measure of at least one of the functions $h_1$ or $h_2$ has support that does not include the point zero. Similarly, the requirement of statement (ii) in theorem 3.3 will certainly be satisfied if the representing measure of at least one of the functions $h_1$ or $h_2$ has compact support.

**Remark 3.5** If, in theorem 3.3, we have a function $q \in \mathcal{Q}_{\mathrm{sym}}$, all integrals with odd powers $k$ on the left-hand side of identity (3.8) are zero due to the symmetry of the measure. Furthermore, for even powers $k$, these integrals may be written as

$$\lim_{\varepsilon \to 0^+} \lim_{y \to 0^+} \frac{2}{\pi} \int_\varepsilon^{\varepsilon^{-1}} x^k \operatorname{Im}\{q(x + iy)\} \, dx. \tag{3.10}$$

**Remark 3.6** Theorem 3.3 cannot be formulated for arbitrary quasi-Herglotz functions. Examples show that it even does not hold for all meromorphic quasi-Herglotz functions, e.g. it can be shown that the quasi-Herglotz function $q(z) := \tan(z) - i$ admits, at $z = \infty$, an asymptotic expansion of order $K = 0$, but there exists no integer $N_\infty$ that would fulfil statement (ii) of theorem 3.3.

# 4. Approximation and optimization based on quasi-Herglotz functions

In this section, we derive the rationale for employing convex optimization as a tool to approximate a given continuous function defined on a compact approximation domain by certain quasi-Herglotz functions. The approximating quasi-Herglotz functions are first restricted to a certain subspace characterized by a particular requirement regarding their Hölder continuity on the approximation domain. Then it is shown that a smaller set of quasi-Herglotz functions generated by finite B-spline expansions (suitable for convex optimization) is dense in the larger space of Hölder continuous quasi-Herglotz functions in the topology induced by any $L^p$-norm. In essence, this development constitutes a straightforward, but very important extension of previous results derived for Herglotz functions [18].

## 4.1. Approximation theory based on quasi-Herglotz functions

To make the statements given above precise, we fix the approximation domain $\Omega \subset \mathbb{R}$ as a finite union of closed and bounded intervals on the real axis.

With a finite B-spline expansion we refer to a finite linear combination of B-splines of any order $m \geq 2$. A B-spline of order $m$ is a compactly supported positive basis spline function which is piecewise polynomial of order $m-1$, i.e. linear, quadratic, cubic, etc., and which is defined by $m+1$ break-points as described, for example, in [37,38]. With a finite uniform B-spline expansion we refer to a finite B-spline expansion with equidistant break-points.

The following definitions and theorems are similar to those in [18] but extended to the current situation with quasi-Herglotz functions. Let $\Omega$ be given as above and let $w > 0$ denote a positive continuous weight function, and let $0 < \alpha < 1$, $1 \leq p \leq \infty$ and $m \geq 2$.

**Definition 4.1** Let $W^{\alpha,p}(w, \Omega) \subset L^p(w, \Omega)$ denote the subspace of all complex-valued functions $q \in C^{0,\alpha}(\Omega)$ with the following property: there exists a quasi-Herglotz function that has a Hölder continuous (with exponent $\alpha$) extension to the closure of some neighborhood $\mathcal{O}$ of $\Omega$ which coincides with $q$ on $\Omega$.

Note that we consider $W^{\alpha,p}(w, \Omega)$ as a subspace of $L^p(w, \Omega)$ and hence equipped with the topology from $L^p(w, \Omega)$.

**Remark 4.2** If it is clear from the context, in the following, we denote by $q$ the quasi-Herglotz function as well as the extension to $\mathbb{C}^+ \cup \mathcal{O}$ and its restriction to $\Omega$.

**Definition 4.3** Let $W^{m,p}(w, \Omega) \subset W^{\alpha,p}(w, \Omega)$ denote the subspace of those functions for which the signed measure $\beta$ (in (2.11)) of the quasi-Herglotz function $q$ in definition 4.1 is absolutely continuous with density $\beta'$ that is a finite uniform B-spline expansion of order $m$.

Note that the sets $W^{\alpha,p}(w, \Omega)$ and $W^{m,p}(w, \Omega)$ are independent of $p$ and $w$, but are equipped with the topology of $L^p(w, \Omega)$.

**Remark 4.4** The signed measure $\beta$ is a not necessarily finite signed Borel measure and can be represented in terms of the finite signed measure $\sigma$ as described in §2.3.

The following theorem is a straightforward generalization of [18, Thm. 3.4] to the situation of quasi-Herglotz functions instead of Herglotz functions.

**Theorem 4.5** *The subspace $W^{m,p}(w, \Omega)$ is dense in $W^{\alpha,p}(w, \Omega)$ with respect to the topology of $L^p(w, \Omega)$.*

*Proof.* Let $\varepsilon > 0$ and let a function $q \in W^{\alpha,p}(w, \Omega)$ be given. Since both the positive and the negative parts of a real-valued Hölder continuous function are again Hölder continuous, it follows that $q$ can be written as $q = h_1 - h_2$ with functions $h_1$ and $h_2$ belonging to the convex cone $V^{\alpha,p}(w, \Omega)$, similar to $W^{\alpha,p}(w, \Omega)$ but generated by extensions of *Herglotz functions* rather than quasi-Herglotz functions. Then, theorem 3.4 in [18, pp. 443–445] implies that there exist functions $\widetilde{h}_1$ and $\widetilde{h}_2$ belonging to the convex cone $W^{m,p}(w, \Omega)$ such that $\|\widetilde{h}_i - h_i\|_{L^p(w,\Omega)} < \varepsilon/2$ for $i = 1, 2$. Hence for $\widetilde{q} := \widetilde{h}_1 - \widetilde{h}_2 \in W^{m,p}(w, \Omega)$ it holds $\|\widetilde{q} - q\|_{L^p(w,\Omega)} < \varepsilon$, which finishes the proof. ∎

**Definition 4.6** Let $F \in C(\Omega)$ and consider the problem to approximate $F$ based on the set of functions $q \in W^{\alpha,p}(w, \Omega)$. The greatest lower bound on the approximation error over the subspace $W^{\alpha,p}(w, \Omega)$ is defined by

$$d := \inf_{q \in W^{\alpha,p}(w,\Omega)} \|q - F\|_{L^p(w,\Omega)}. \tag{4.1}$$

Note that the distance $d$ depends on the chosen topology of $L^p(w, \Omega)$, but is independent of the Hölder exponent $\alpha$ (cf. [18]). The following theorem demonstrates the usefulness of employing finite B-spline expansions in the associated approximation problem.

**Theorem 4.7** *The greatest lower bound on the approximation error defined in* (4.1) *is given by*

$$d = \inf_{q \in W^{m,p}(w,\Omega)} \|q - F\|_{L^p(w,\Omega)}. \tag{4.2}$$

The theorem is a straightforward consequence of theorem 4.5 together with an application of the triangle inequality. It is noted that the distance $d$ is independent of the Hölder exponent $\alpha$ as well as of the spline order $m$ (cf. [18]). The following obvious corollary can be used when the measure of the approximating quasi-Herglotz function contains a set of point masses. Such cases will be discussed in §§4.2 and 5.

**Corollary 4.8** *Let* $W(w, \Omega) \subset W^{\alpha,p}(w, \Omega)$ *be a set which contains* $W^{m,p}(w, \Omega)$. *Then,* $W(w, \Omega)$ *is dense in* $W^{\alpha,p}(w, \Omega)$ *and it holds that*

$$d = \inf_{q \in W(w,\Omega)} \|q - F\|_{L^p(w,\Omega)}. \tag{4.3}$$

## 4.2. Convex optimization with B-splines

The significance of the theorems 4.5 and 4.7 is that B-spline expansions [37–39], which are well suited for numerical optimization [18,40,41], can be used to approximate a given continuous function $F$ with arbitrary small deviation from the greatest lower bound defined in (4.1). A detailed description of the associated convex optimization problem is given as follows.

Let the approximation domain $\Omega$, the target function $F \in C(\Omega)$ and the weight function $w \in C(\Omega)$ be given as above, and let $0 < \alpha < 1$, $1 \leq p \leq \infty$ and $m \geq 2$. As approximating functions we can use functions $q$ from a set $W(w, \Omega)$ defined by the following representations:

$$q(x) = a + bx + \sum_{i=1}^{M} \frac{p_i}{\xi_i - x} + \text{p.v.} \int_{-\infty}^{\infty} \left( \frac{1}{\xi - x} - \frac{\xi}{1 + \xi^2} \right) \beta'(\xi) \, d\xi + i\pi\beta'(x) \tag{4.4}$$

$$= \breve{a} + bx + \sum_{i=1}^{M} \frac{p_i}{\xi_i - x} + \text{p.v.} \int_{-\infty}^{\infty} \frac{1}{\xi - x} \beta'(\xi) \, d\xi + i\pi\beta'(x) \tag{4.5}$$

for $x \in \Omega$, and where the second part of the integral in (4.4) has been absorbed into the constant $\breve{a}$ in (4.5). In (4.4) and (4.5) the density $\beta'$ is a finite uniform B-spline expansion as in definition 4.3, and a finite number of point masses at $\xi_i \notin \Omega$ with real-valued amplitudes $p_i$, $i = 1, \ldots, M$, have also been included. It is noted that the set $W(w, \Omega)$ satisfies the condition $W^{m,p}(w, \Omega) \subset W(w, \Omega) \subset W^{\alpha,p}(w, \Omega)$ of corollary 4.8.

In particular, we employ here B-spline basis functions $p_n(x)$ of fixed polynomial order $m - 1$ for $n = 1, \ldots, N$, where $N$ is the number of B-splines, and $\hat{p}_n(x)$ the (negative) Hilbert transform [29] of the B-spline functions. Explicit formulae for general uniform as well as non-uniform B-splines and their Hilbert transforms are given in [42, Sec. 3.1]. Let $q_N \in W$ denote approximating functions represented as in (4.5), and hence

$$\text{Im}\{q_N(x)\} = \pi\beta'(x) = \sum_{n=1}^{N} c_n p_n(x) \tag{4.6}$$

and

$$\text{Re}\{q_N(x)\} = \breve{a} + bx + \sum_{i=1}^{M} \frac{p_i}{\xi_i - x} + \sum_{n=1}^{N} c_n \hat{p}_n(x), \tag{4.7}$$

for $x \in \Omega$, and where $c_n$ are the corresponding B-spline expansion coefficients. Note that all the parameters $\breve{a}$, $b$, $\{p_i\}_{i=1}^{M}$ and $\{c_n\}_{n=1}^{N}$, as well as the break-points of the B-splines defined above depend on $N$. It is further assumed that the support of $q_N$ grows with $N$ at the same time as the distance $\delta$ between breakpoints decreases, e.g. as $|\text{supp}\{q_N\}| = \sqrt{N}$ and $\delta = |\text{supp}\{q_N\}|/N = 1/\sqrt{N}$. For a fixed $N$, the minimization of the norm of the approximation error $\|q_N - F\|_{L^p(w,\Omega)}$ is a finite-dimensional convex optimization problem over the real parameters $\breve{a}$, $b$, $p_i$ and $c_n$, and we denote the optimal value $d_N$. The important implication of theorem 4.7 and corollary 4.8 is that $d_N \to d$ as $N \to \infty$. Finally, it is noted

that for a numerical implementation using, for example, the CVX MATLAB software for disciplined convex programming [41] the calculation of the norm above must be approximated based on a finite set of sample points in $\Omega$. However, due to the uniform continuity of all functions involved, this can, in principle, be done within arbitrary numerical accuracy.

Now that we have established the rationale for using numerical convex optimization as a tool for approximating a given continuous function based on the set of quasi-Herglotz functions, we can also expand the setting by incorporating any additional convex constraints of interest (see also [17,43,44]). For example, we can include upper and lower bounds on the density $\beta'$ stated as

$$
\begin{aligned}
& \text{minimize} \quad \|q - F\|_{\mathrm{L}^p(w,\Omega)} \\
& \text{subject to} \quad b_{\mathrm{lower}}(x) \leq \beta'(x) \leq b_{\mathrm{upper}}(x),
\end{aligned}
\tag{4.8}
$$

where the optimization is over $q \in W(w, \Omega)$ and $b_{\mathrm{lower}}$ and $b_{\mathrm{upper}}$ are suitable functions. Note that these functions can be used for constraining the density $\beta'(x)$ outside of $\Omega$ to prevent non-physical oscillatory behaviour of the resulting function outside the approximation domain. Also, these constraints are useful in regularization of the low-frequency behaviour of materials; see the numerical examples in §5. In practice, this might for instance amount to solving for $q_N \in W(w, \Omega)$

$$
\begin{aligned}
& \text{minimize} \quad \|q_N - F\|_{\mathrm{L}^p(w,\Omega)} \\
& \text{subject to} \quad \theta_{\mathrm{lower},j} \leq \theta_j \leq \theta_{\mathrm{upper},j}, j \in J,
\end{aligned}
\tag{4.9}
$$

where $N$ is fixed, $J$ is a finite index set and the vector $\theta$ may consist of any of the parameters $\theta_j \in \{\check{a}, b, p_1, \ldots, p_M, c_1, \ldots, c_N\}$, for $j \in J$.

When *a priori* information is available about the asymptotic properties of a given non-passive system to be approximated and which admits the sum rules discussed in §3, the identities (3.8) can be involved in an optimization (4.8) as an additional convex constraint. Due to the finite-dimensional approximation (4.6), the left-hand side of (3.8) becomes

$$
\lim_{\varepsilon \to 0^+} \lim_{y \to 0^+} \frac{1}{\pi} \int_{\varepsilon < |x| < (1/\varepsilon)} x^k \operatorname{Im}\{q(x + iy)\} \, dx = \lim_{\varepsilon \to 0^+} \frac{1}{\pi} \sum_{n=1}^{N} c_n \int_{\varepsilon < |x| < (1/\varepsilon)} x^k p_n(x) \, dx,
\tag{4.10}
$$

for even $k = -2N_0, \ldots, 2N_\infty$, see theorem 3.3, and which can be employed as an additional constraint in the optimization formulation (4.9).

# 5. Numerical examples

In the numerical examples presented below, non-passive approximation is employed as a tool to determine optimal realizations (in the sense of a mathematical representation) of non-passive systems with a given target response over the approximation domain. The target functions to be approximated are symmetric, and thus we employ symmetric quasi-Herglotz functions to solve the convex optimization problems.

The symmetry property (2.2) implies that the representation based on (4.6) and (4.7) can be simplified as

$$
\operatorname{Re}\{q_N(x)\} = bx + \frac{p_0}{-x} + \sum_{i=1}^{M} p_i \left( \frac{1}{\xi_i - x} - \frac{1}{\xi_i + x} \right) + \sum_{n=1}^{N} c_n [\hat{p}_n(x) - \hat{p}_n(-x)]
\tag{5.1}
$$

and

$$
\operatorname{Im}\{q_N(x)\} = \sum_{n=1}^{N} c_n [p_n(x) + p_n(-x)],
\tag{5.2}
$$

respectively, where $\check{a} = 0$ and $p_0$ denotes the amplitude of the point mass located at 0. In connection with the optimization formulation (4.8) and (4.9) established above, it is also convenient here to introduce the notation $\Omega_{\mathrm{opt}} = I_1 \cup I_2$, where $\Omega_{\mathrm{opt}}$ is the optimization domain consisting of two disjoint sets $I_1$ and $I_2$ where the approximating measure is required to be non-negative ($p_i \geq 0$ or $c_n \geq 0$) and non-positive ($p_i \leq 0$ or $c_n \leq 0$), respectively. Hence, the support of the measure is contained in $\Omega_{\mathrm{opt}}$.

The approximation methods that we have described above are general, and can be applied to any quasi-Herglotz function and for a range of physics and engineering applications. In the examples below, we consider a sequence of interesting different optimization constraints that we think are generally applicable. In particular, we consider different constraints on the optimized function outside

the approximation domain, $\Omega$, where passivity or conditions of non-passivity can be applied. In the first numerical example presented in §5.1, we consider a target response, $F$, which is the restriction of a non-Herglotz function, here $F = -h_0$, where $h_0$ is the $\Omega$ restriction of a Herglotz function. In the second example described in §5.2, we use the same target response; however, we apply the non-passive approximation framework to determine an optimal realization of the system. Here, we consider a constrained amplifying region over a fixed bandwidth outside the approximation domain. In addition, we study the dependence of the approximation error on the size of the approximation domain for a given fixed optimization domain. The third example in §5.3 is focused on the non-passive approximation of a given system, where the approximating quasi-Herglotz function is generated by a measure consisting of point masses. In the fourth numerical example given in §5.4, we are interested to determine an optimal non-passive realization of the given system with additional constraints of its asymptotic properties, i.e. the behaviour in the small- and large-argument limits. Hereby, we extend the convex optimization problem (4.9) with an additional sum-rule constraint which is based on (4.10).

Although the developed theory is generally applicable, in the following examples, we select a canonical electromagnetic application, which is the modelling of permittivity functions that characterize metamaterials with desired exotic properties (fixed negative permittivity, which is of interest in, for example, plasmonic applications). For the non-passive cases, the optimized dielectric permittivity functions $\epsilon_{opt}$ can have negative imaginary part over some frequency intervals. Note that the functions we study here correspond to linear and stable systems. Consequently, these functions have no poles in the upper-half complex plane. However, these functions can also be used as an input to another system, e.g. the transmission or reflection coefficients [45] from a dielectric slab, and cause an instability of the resulting system. In practice, any instability issues associated with non-passive systems will always be limited by other external factors such as the saturation of the gain media [21], which is not considered in this paper.

Here, the independent variable is a dimensionless real-valued normalized frequency $x$ corresponding to an angular frequency $\omega$ in rad/s. For simplicity and since the approximants $q(x)$ and $q_N(x)$ in (4.8) and (4.9) are conjugate symmetric on $\mathbb{R}$, in particular, $\text{Im}\{q_N\}$ is an even function when restricted to $\mathbb{R}$, we will only specify and visualize the right side of the approximation domain, i.e. $\Omega \cap \mathbb{R}_+$.

## 5.1. Passive approximation of a system with a given target response

An interesting canonical example for which (4.1) gives a non-trivial bound is with the passive approximation of a negative symmetric Herglotz function $F = -h_0$, which can be Hölder continuously extended to $\mathbb{C}^+ \cup \Omega$, and which has the large-argument asymptotics $h_0(z) = b_1^0 z + o(z)$ as $z \hat{\rightarrow} \infty$. Based on the theory of Herglotz functions and associated sum rules [6], it can be shown that

$$\|h - F\|_\infty \geq (b_1 + b_1^0)\frac{1}{2}|\Omega|, \tag{5.3}$$

for all Herglotz functions $h$ with large-argument asymptotics $h(z) = b_1 z + o(z)$ as $z \hat{\rightarrow} \infty$ (see [10,18]). Here, $|\Omega|$ is the length of the interval $\Omega$.

As an application, consider a passive approximation of a metamaterial as in [10], and note that the case with passive systems and passive approximation based on Herglotz functions as in [18] constitutes a special case of the non-passive approximation based on the representations (5.1) and (5.2), with $b_{\text{lower}} = 0$ in (4.8), i.e. $\beta'(x) \geq 0$ for all $x$.

For a passive metamaterial, a dielectric permittivity function $\epsilon(z)$ is considered, where $h(z) = z\epsilon(z)$ is the associated symmetric Herglotz function [10]. The high-frequency permittivity of the metamaterial is assumed to be given by $\epsilon_\infty$, and hence $b_1 = \epsilon_\infty$. A real-valued and constant target permittivity $\epsilon_t < 0$ is given over the approximation interval $\Omega$ defined by $\Omega = [1 - B/2, 1 + B/2]$, where $B$ is the relative bandwidth, $0 < B < 2$, and hence $F(x) = x\epsilon_t$ with $h_0(z) = -z\epsilon_t$, and $b_1^0 = -\epsilon_t$. Now, by using (5.3) the resulting physical bound can thus be obtained as (see also [10,18])

$$\|\epsilon - \epsilon_t\|_\infty = \|h - F\|_{L^\infty(w, \Omega)} \geq \frac{\|h - F\|_\infty}{1 + B/2} \geq \frac{(\epsilon_\infty - \epsilon_t)B}{2 + B} =: \Delta. \tag{5.4}$$

Note that here $\|\epsilon - \epsilon_t\|_\infty = \|h - F\|_{L^\infty(w, \Omega)}$, where the weight function is $w(x) = 1/x$ for $x \in \Omega$, with $0 \notin \Omega$, and where $\Delta$ is the resulting physical bound.

Let the target function $F = x\epsilon_t$ be defined over the approximation domain $\Omega = [1 - B/2, 1 + B/2]$, where the relative bandwidth $B = 0.02$, and let the support of the generating measure $\text{supp}\{\beta\} \cap \mathbb{R}_+$ be contained in the optimization domain $\Omega_{opt} = \{0\} \cup [0.97, 1.03]$ including one positive point mass with amplitude $p_0$ at the origin, and the density $\beta'$ is constrained to be non-negative, i.e. $\beta'(x) \geq 0$.

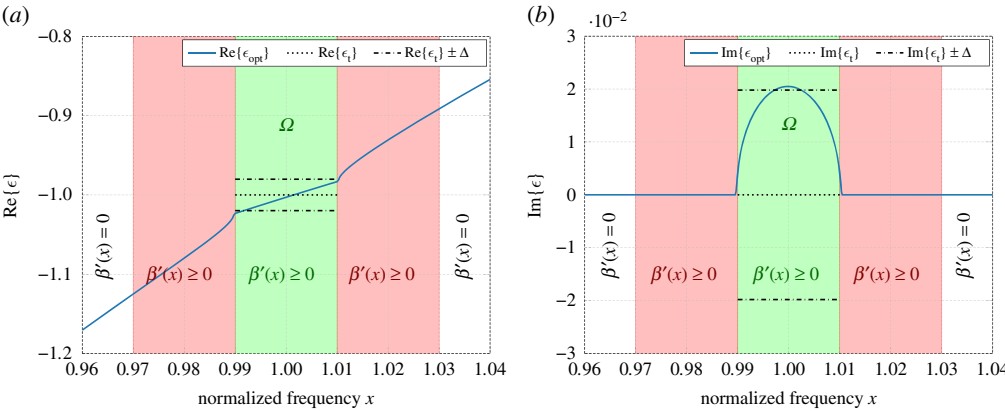

**Figure 1.** Real and imaginary part of the optimal passive permittivity function approximating a metamaterial with given $\epsilon_t = -1$ over the approximation domain $\Omega$. The corresponding sum-rule-bound limits based on (5.4) are given by $\epsilon_t \pm \Delta$. (a) Real part of permittivity. (b) Imaginary part of permittivity.

Figure 1 shows the result of optimization (4.9) carried out using $N = 100$ uniform linear B-splines (of order $m = 2$) for given parameters $\epsilon_t = -1$ and $\epsilon_\infty = 1$. The real and imaginary parts of the resulting permittivity function $\epsilon_{\text{opt}}$ are shown in figure 1a and 1b, respectively, including comparison with the fundamental sum-rule-bound limits $\epsilon_t \pm \Delta$, when $\Delta$ is given by (5.4). Interestingly, the optimized function is, in principle, supported only on $\Omega$, i.e. the optimal solution for $\beta'$ is approximately zero outside the approximation domain $\Omega$, except for the point mass with $p_0 \approx 79.1$. It should be noted that the point mass at the origin contributes with a response at frequencies $x \neq 0$, which is very similar to that of a Drude model with sufficiently large relaxation time $\tau$, so that $x\tau \gg 1$.

## 5.2. Non-passive approximation of a system with a given target response

Consider an approximation problem, similar to the one described in §5.1, with the difference that the approximating functions are not restricted to Herglotz functions, but to quasi-Herglotz functions. Let $\Omega_{\text{opt}} = I_1 \cup I_2$ denote the domain of optimization of the measure $\beta$, where $I_2 = [0.97, 0.99) \cup (1.01, 1.03]$ is the frequency set where the density $\beta'$ is constrained to be non-positive, i.e. $\beta'(x) \leq 0$.

Figure 2 shows the corresponding optimization results using $I_1 = \{0\}$ including only a point mass with amplitude $p_0$ at the origin. It has been observed that by letting $I_1 = \{0\} \cup \Omega$ and letting $\beta'(x) \geq 0$ over $\Omega$, one can obtain negligible deviation from the optimal solution presented in figure 2. In this optimization, we used $N = 100$ linear B-splines, which is sufficient to achieve an approximation error of order of magnitude $10^{-6}$. Here, the resulting point mass has a magnitude $p_0 \approx 79.1$, which is essentially the same as in the example presented in §5.1, but the support of the approximating function $q_N$ over $I_2$ becomes concentrated to the outermost endpoints of the set. This seems to suggest that two point masses with amplitudes $p_1$ and $p_2$ located at the two outermost endpoints $x_1 \approx 0.971$ (lower endpoint) and $x_u \approx 1.029$ (upper endpoint) of $I_2$ should be sufficient for the optimization, where the approximating function has the representation based on (5.1). Hence, figure 2 also includes optimization results, where the measure $\beta$ consists solely of these two point masses with negative amplitudes $p_1$ and $p_2$, together with the original point mass at 0 with amplitude $p_0$. Figure 2b also shows the optimized point masses with $p_1 \approx -8.7$ and $p_2 \approx -8.48$ (indicated by the dark-red 'o') normalized to the same area as the corresponding linear B-spline basis functions. The example illustrates that it is possible under certain circumstances to obtain a much better realization of the target permittivity $\epsilon_t$ given over the approximation interval with smaller approximation error as a non-passive system by using quasi-Herglotz functions rather than by using only Herglotz functions; compare the results in figures 2a and 1a, respectively.

Now, let $\Omega_{\text{opt}} = I_1 \cup I_2$, where $I_1 = \{0\}$ (where $p_0 \geq 0$), and $I_2 = [0.97, 1 - B/2) \cup (1 + B/2, 1.03]$ (where $\beta'(x) \leq 0$). The approximation domain is $\Omega = [1 - B/2, 1 + B/2]$. Figure 3 illustrates how the size of the approximation domain $|\Omega| = B$ affects the optimal realization of the desired system response. Here, the support of the measure $q_N$ is concentrated at the outermost frequencies of the set $I_2$ as demonstrated in the example above. In figure 3a is shown optimal realizations $\text{Re}\{\epsilon_{\text{opt}}\}$ of the desired target function $\epsilon_t = -1$ for two different sizes of $\Omega$, where $B = 0.02$ and $B = 0.056$, respectively, and a comparison with the passivity bounds $\text{Re}\{\epsilon_t\} \pm \Delta$ defined in (5.4).

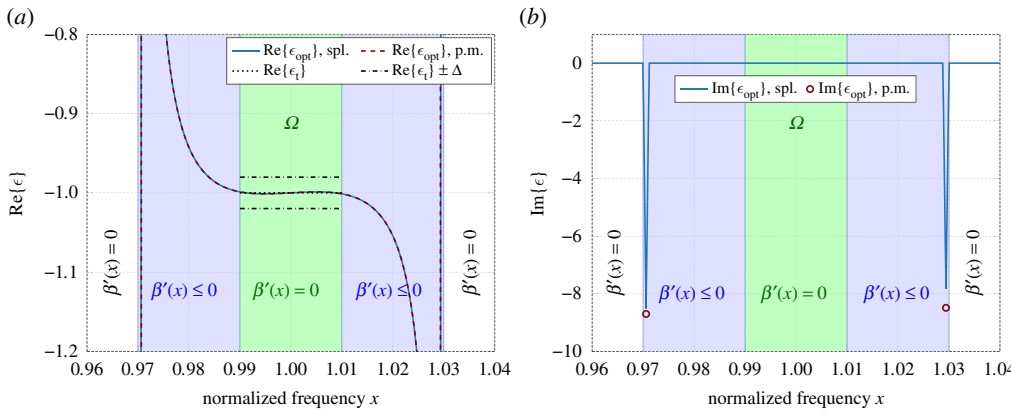

**Figure 2.** Real and imaginary parts of the optimal non-passive permittivity function approximating a metamaterial with given, negative target permittivity $\epsilon_t = -1$ over the approximation domain $\Omega$. The corresponding sum-rule-bound limits based on (5.4) are given by $\epsilon_t \pm \Delta$. Two cases are considered, a spline approximation in $l_2$ (blue) and the point mass approximation in $l_2$ (red). (a) Real part of permittivity. (b) Imaginary part of permittivity.

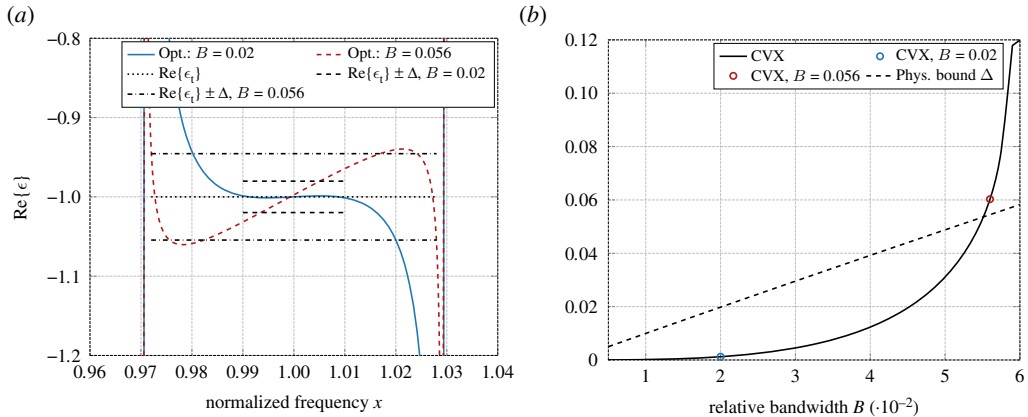

**Figure 3.** (a) Real part of the non-passive permittivity function approximating a metamaterial with given $\epsilon_t = -1$ over the approximation domain $\Omega = [1 - B/2, 1 + B/2]$; (b) Approximation error as a function of relative bandwidth B. (a) Real part of permittivity. (b) Approximation error $\|\epsilon - \epsilon_t\|_\infty$.

Figure 3b shows how the approximation error $\|\epsilon - \epsilon_t\|_\infty$ depends on the size of the approximation domain given as a function of the relative bandwidth B, i.e. $\Omega = [1 - B/2, 1 + B/2]$. From this figure it can be concluded that by increasing the size of $\Omega$ towards the outermost points of $I_2$, the approximation error increases, and for high values of B the optimization results become even worse than the results based on Herglotz functions for the passive case.

## 5.3. Optimization with point masses

Consider again the optimization based solely on point masses as in §5.2 above. For this problem, let $\Omega_{opt} = I_1 \cup I_2$, where $I_1 = \{0\}$ and $I_2 = \{x_l\} \cup \{x_u\}$ denote the domain of optimization of the measure $\beta$, and $\Omega = [1 - B/2, 1 + B/2]$ the approximation domain. Here, the approximating quasi-Herglotz function has the representation based on (5.1) with $c_n = 0$, $x_l$ and $x_u$ are the lower and upper normalized frequencies, where the point masses with negative amplitudes are located. Note that the optimization is done solely over the three point masses with amplitudes $p_0$ (which is located at the origin), $p_1$ and $p_2$.

Figure 4 shows the optimization results and how the approximation error $\|\epsilon - \epsilon_t\|_\infty$ depends on the location of the assumed *a priori* point mass, where $\epsilon_t = -1$ is the given target permittivity function. From figure 4b, it can be concluded that the non-passive approximation based on point masses provides a good agreement between the target function and the optimal solution based on the approximating quasi-Herglotz function generated by point masses. In particular, the approximation error has an L-curve

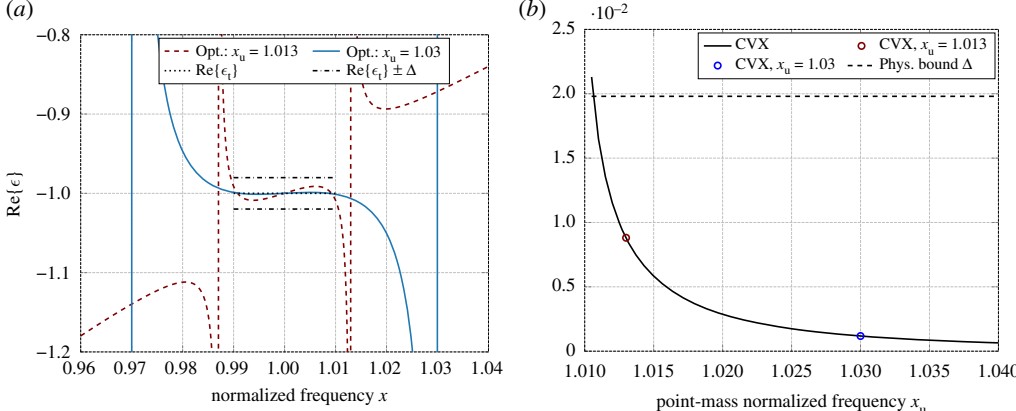

**Figure 4.** (a) Real part of the non-passive permittivity function optimized with point masses only. The corresponding sum-rule-bound limits based on (5.4) are given by $\epsilon_t \pm \Delta$; (b) Approximation error as a function of upper point mass normalized frequency $x_u$ (the normalized frequency of lower point mass is $x_l = 1 - (x_u - 1)$). (a) Real part of permittivity. (b) Approximation error $\|\epsilon - \epsilon_t\|_\infty$.

characterization with transition at about 2% from the normalized frequency $x_c$; see the approximation error and the optimal realizations in figure 4b and 4a, respectively.

## 5.4. Optimization with sum-rule constraints

Now, we would like to further constrain the optimization (4.8) to determine an optimal realization of a system with additionally given small- and large-argument asymptotic properties. Consequently, for the given target function $F$, the convex optimization problem (4.9) is modified with an additional convex constraint obtained from (4.10) for $k = -2$ as

$$\lim_{\varepsilon \to 0^+} \frac{1}{\pi} \int_{\varepsilon < |x| < (1/\varepsilon)} \frac{\text{Im}\{q(x)\}}{x^2} \, dx = a_1 - b_1, \tag{5.5}$$

where $a_1$ and $b_1$ denote the expansion coefficients of the small- and the large-argument asymptotic expansion of the given system response, respectively. Note that $b_1$ in constraint (5.5) coincides with $b$ in the representation (4.5) of the approximating quasi-Herglotz function.

As an application, modelling of a permittivity function is considered here, where the target permittivity $\epsilon_t = -1$ is fixed over the approximation domain $\Omega = [1 - B/2, 1 + B/2]$, $0 < B < 2$. The small- and the large-argument asymptotics of this system are represented by the static and the high-frequency permittivities, i.e. $a_1 = \epsilon_s$ and $b_1 = \epsilon_\infty$, respectively.

Let $\Omega_{\text{opt}} = I_1 \cup I_2$ denote the domain of optimization of $\beta'$, $I_1 = [0.01, 0.9] \cup (1.5, 2]$ the frequency set, where the density is restricted to be non-negative, i.e. $\beta'(x) \geq 0$, and $I_2 = (1.1, 1.5)$ the frequency set, where the density $\beta'(x) \leq 0$. For this optimization, the relative bandwidth $B = 0.2$ (and hence $\Omega = [0.9, 1.1]$), the asymptotic constraints are $a_1 = \epsilon_s = 3$ and $b_1 = \epsilon_\infty = 1$, and $N = 1000$ linear B-splines are used due to the increased size of $\Omega_{\text{opt}}$, which is sufficient for an accurate solution. Further, the set $I_1$ has been increased in comparison with the previous example to control the realization of the optimal solution with the desired low-argument asymptotic behaviour.

Figure 5a,b,c depicts the corresponding optimization results with no *a priori* point masses. The obtained optimization result shows a good agreement of the target function $\epsilon_t = -1$ over the approximation domain $\Omega$; see figure 5b. The approximation error $\|\epsilon - \epsilon_t\|_\infty$ is much less than the physical bound for passive metamaterials (5.4), and hence the optimal solution $\epsilon_{\text{opt}}$ fits well $\epsilon_t$ over $\Omega$. Also, it is reassuring to note that the optimal solution satisfies the asymptotic requirement for the small-argument limit, i.e. $\epsilon_{\text{opt}} \to 3$ as $x \to 0$; see figure 5a. We have observed that the same result can be accurately achieved when the measure $\beta$ consists of two point masses with amplitudes $p_1 \approx 989.9$ and $p_2 \approx -790.8$ placed at $x_1 \approx 0.469$ and $x_u \approx 1.499$, respectively, where $x_u$ is the upper outermost frequency of the non-passive region represented by the set $I_2$. In figure 5d, the approximation error is shown as a function of the small-argument asymptotic constraint $a_1 = \epsilon_s$. It is interesting to note that the approximation error decreases as $\epsilon_s$ increases, and meanwhile, the location of the point mass with positive amplitude $p_1$ moves towards the origin. Hence, in the limit, when the point mass approaches zero, we obtain a result which is very similar to the non-passive approximation case described in §§5.2 and 5.3.

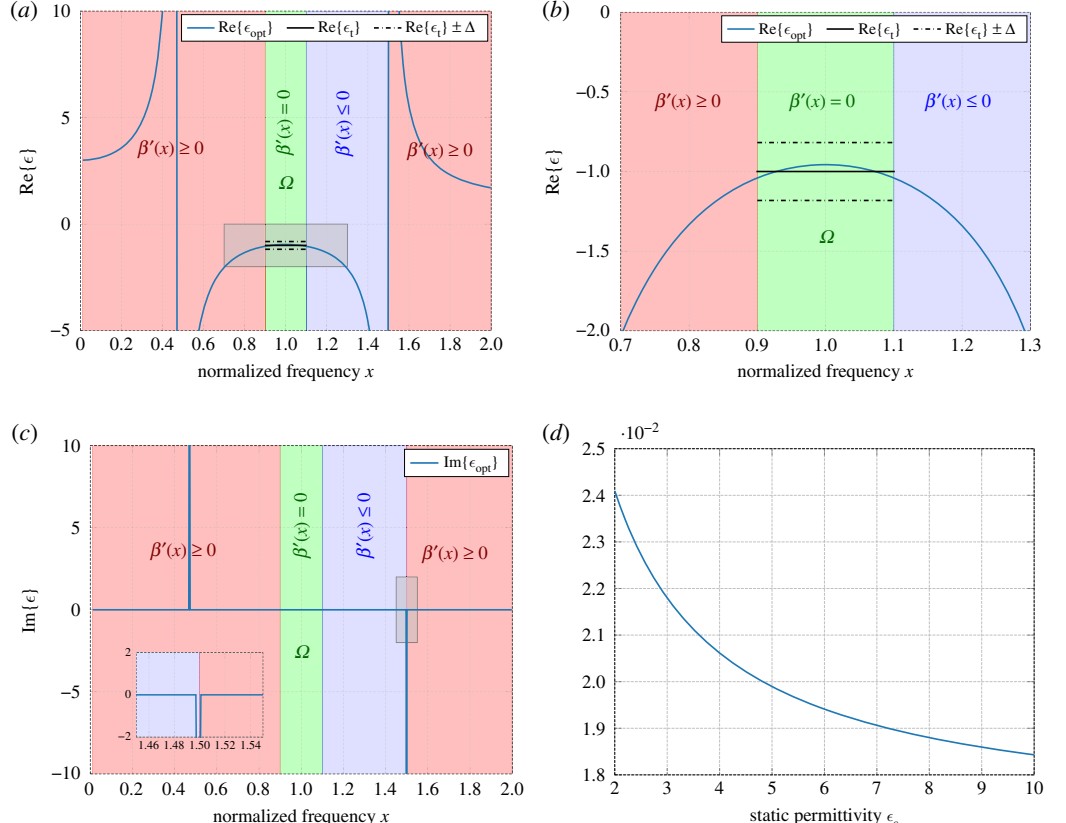

**Figure 5.** ($a$–$c$) Real and imaginary parts of the optimal non-passive permittivity function approximating a metamaterial with $\epsilon_t = -1$, $\epsilon_s = 3$ and $\epsilon_\infty = 1$. The zoomed-in region in ($a$) is depicted in ($b$). The corresponding sum-rule-bound limits based on (5.4) are given by $\epsilon_t \pm \Delta$; ($d$) Approximation error plotted as a function of the static permittivity $\epsilon_s$. ($a$) Real part of permittivity. ($b$) Real part of permittivity (zoom). ($c$) Imaginary part of permittivity. ($d$) Approximation error $\|\epsilon - \epsilon_t\|_\infty$.

# 6. Conclusions

In this paper, the non-passive framework for a certain class of non-passive causal systems has been formulated. This has been done by extending the existing class of Herglotz functions to the class of quasi-Herglotz functions, which is obtained by taking all possible differences of two Herglotz functions. Based on the integral representation formulae for Herglotz functions using finite measures, we have shown that quasi-Herglotz functions can be described by an integral representation formula using signed Borel measures. For Herglotz functions, one can also use an equivalent, possibly non-finite measure, in their representation formula. However, this is not the case for quasi-Herglotz functions when the measure is non-finite where only some functions admit integral representations via non-finite signed measures. Quasi-Herglotz functions can also be analytically extended to some interval of the real axis in the same way as Herglotz functions, provided the density of measure of the function is Hölder continuous on some open neighbourhood of this interval, which is important for the non-passive framework. Furthermore, we show that quasi-Herglotz functions admit, under certain additional constraints, sum-rule identities that generalize the known identities for Herglotz functions, and which allow us to control small- and large-argument asymptotics of desired non-passive systems in optimization problems.

We have also demonstrated that a family of B-splines can be used in the representation of approximating quasi-Herglotz functions, which is used in a number of numerical examples. It has been concluded that a very efficient mathematical representation of a non-passive metamaterial with $\epsilon_t \approx -1$ (which is typical in plasmonic applications) can be achieved by choosing point masses representing the power excitation at certain frequencies outside of the approximation domain. A further constrained problem for non-passive metamaterials with controlled low- and high-frequency responses shows that the sum-rule identities can be efficiently used in the realization of such permittivities with desired properties as a constraint for the convex optimization problem (4.8).

Data accessibility. The paper has no experimental data. The numerical simulations were carried out using the open-source CVX MATLAB package [41]. All of the data needed to run the simulations are specified in the article.

Authors' contributions. All authors have contributed to the writing of the manuscript, to the development of the mathematical model and to the analysis of the results. Additionally, the numerical simulations were performed by Y.I. All authors have given final approval for publication.

Competing interests. The authors declare that they have no competing interests.

Funding. This work was funded by the Swedish Foundation for Strategic Research (SSF), grant no. AM13-0011 under the programme Applied Mathematics, and the project Complex analysis and convex optimization for EM design.

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
