## [Reviewer comments · Royal Society Open Science]

Review History

RSOS-191541.R0 (Original submission)

Review form: Reviewer 1

Is the manuscript scientifically sound in its present form?

Yes

Are the interpretations and conclusions justified by the results?

Yes

Is the language acceptable?

Yes

Do you have any ethical concerns with this paper?

No

Have you any concerns about statistical analyses in this paper?

No

Recommendation?

Accept as is

Comments to the Author(s)

The authors have satisfactorily responded to my comments and I recommend publication. It is a nice advance on the huge literature on the analytic properties of the response functions of passive systems, extended here to active systems.

Review form: Reviewer 2

Is the manuscript scientifically sound in its present form?

Yes

Are the interpretations and conclusions justified by the results?

Yes

Is the language acceptable?

Yes

Do you have any ethical concerns with this paper?

No

Have you any concerns about statistical analyses in this paper?

No

Recommendation?

Accept as is

Comments to the Author(s)

This version is an improvement upon the previous submission, and I am happy to recommend its publication.

Decision letter (RSOS-191541.R0)

22-Nov-2019

Dear Mr Ivanenko:

It is a pleasure to accept your manuscript entitled "Quasi-Herglotz functions and convex optimization" in its current form for publication in Royal Society Open Science. The comments of the reviewer(s) who reviewed your manuscript are included at the foot of this letter.

You can expect to receive a proof of your article in the near future. Please contact the editorial

office (openscience_proofs@royalsociety.org) and the production office (openscience@royalsociety.org) to let us know if you are likely to be away from e-mail contact -- if you are going to be away, please nominate a co-author (if available) to manage the proofing process, and ensure they are copied into your email to the journal.

Kind regards,
Lianne Parkhouse
Editorial Coordinator
Royal Society Open Science
openscience@royalsociety.org

on behalf of the Associate Editor, and Professor Professor Mark Chaplain (Subject Editor).

Reviewer(s)' Comments to Author:

Reviewer: 1
Comments to the Author(s)

The authors have satisfactorily responded to my comments and I recommend publication. It is a nice advance on the huge literature on the analytic properties of the response functions of passive systems, extended here to active systems.

Reviewer: 2
Comments to the Author(s)

This version is an improvement upon the previous submission, and I am happy to recommend its publication.

Follow Royal Society Publishing on Twitter: [@RSocPublishing](https://twitter.com/RSocPublishing)
Follow Royal Society Publishing on Facebook:
<https://www.facebook.com/RoyalSocietyPublishing.FanPage/>
Read Royal Society Publishing's blog: <https://blogs.royalsociety.org/publishing/>